# Improvement in Trapezoidal Pulse Shaping Pile-Up in Nuclear Signal Processing

Junlong Wu [1,2,*] and Xianguo Tuo [1,2,*]

1    College of Nuclear Technology and Automation Engineering, Chengdu University of Technology, Chengdu 610059, China

2    Fundamental Science on Nuclear Waste and Environment Security Laboratory, Southwest University of Science and Technology, Mianyang 621010, China

*    Correspondence: wujunlong@swust.edu.cn (J.W.); tuoxianguo@suse.edu.cn (X.T.)

**Abstract:** In digital nuclear spectroscopy, trapezoidal shaping is widely used. Compared with traditional CR-RC4 semi-Gaussian shaping, it has a better energy resolution and higher counting rates, but does not void the pulse pile-up in the case of extreme counting rates. In this paper, a new recursive algorithm is proposed that can improve the anti-pile-up ability, and is easy to implement in any DSP-based processor that is used in any digital pulse shaping filter section. The complete deduction and simulation are presented in this paper.

**Keywords:** trapezoidal pulse shaping; anti-pile-up; new trapezoidal pulse algorithm

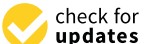



## 1. Introduction

In 1972, Radeka et al. used a digital filter to process the nuclear signal at the Brookhaven National Laboratory [1]. At the time, due to the limitations associated with the improvement of the hardware, it was hard to implement. Subsequently, different experts and scholars have studied how technology can be used to improve spectrometer performance. In recent years, digital technology has made great progress in the development of nuclear instruments. Digital pulse shaping algorithms are an important element of the digital spectrometer, and are becoming easier to implement using the field programmable gate array (FPGA). Trapezoidal pulse shaping algorithms are widely applied to the digital spectrometer because of their almost optimum signal-to-noise ratio (SNR). Jordanov et al. developed the recursive algorithm for trapezoidal pulse shaping using the convolution method [2–6]. However, pulse pile-up is an issue that cannot be resolved in the trapezoidal pulse shaping algorithm when applied to high rates. Hongxu et al. only gave the relationship between the shaping parameters of the trapezoidal pulse shaping algorithm and the trapezoidal pulse shape [7,8]. Jie yu et al. proposed a method to reject the pile-up pulse continuous zone when a better approach is not available to proceed with the trapezoidal pulse pile-up [9].

In this paper, we provide a new method for improving the trapezoidal shaping and its performance against pile-up. The simulation results show that this method can remove and improve the pile-up performance in trapezoidal pulse shaping.

## 2. The Pile-Up of the Trapezoidal Shaping Algorithm

### 2.1. Detector Output Signal Representation

The detector output of the single signal can be expressed as the step signal in Equation (1):

$$u(t) = \begin{cases} 1 \ldots\ldots, t > 0 \\ 0 \ldots\ldots, else \end{cases} \tag{1}$$

Figure 1 shows the detector output of the single signal in the time domain. It can be considered to be a unit step signal. Moreover, the continuous time sequence of the detector outputs can be represented as the added step signals in Equation (2):

$$x(t) = \sum_i u(t - t_i) \tag{2}$$

where $t_i$ follows the Poisson distribution. Because of the arbitrary adjacent pulse time interval, which can be either short or long, and mainly depends on the position of the ionization chamber, the time interval will show a wide range from us to ms. Hence, we hope the next stage of the time constant of the charge sensitive amplifier (CSA) will be matched appropriately. In this way, it is easy to reduce pile -up in the CSA output. However, this will not prevent the occurrence of pile-up.

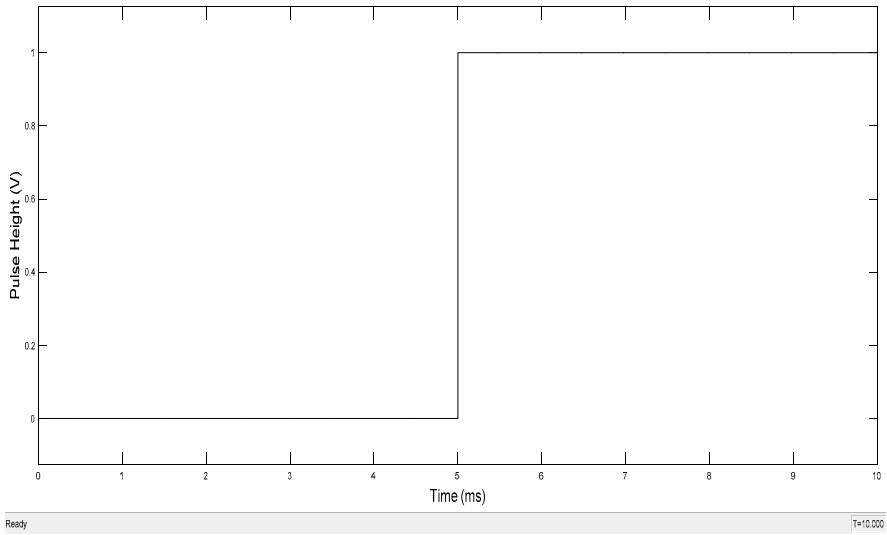

**Figure 1.** Detector equivalent output $u(t)$.

Figure 2 shows the continuous time sequence of the detector output without the discharge circuits. The time domain can be used to describe the staircase sequence signal and the time interval between the two pulses following the Poisson distribution. This indicates that the time interval may be very short in some cases.

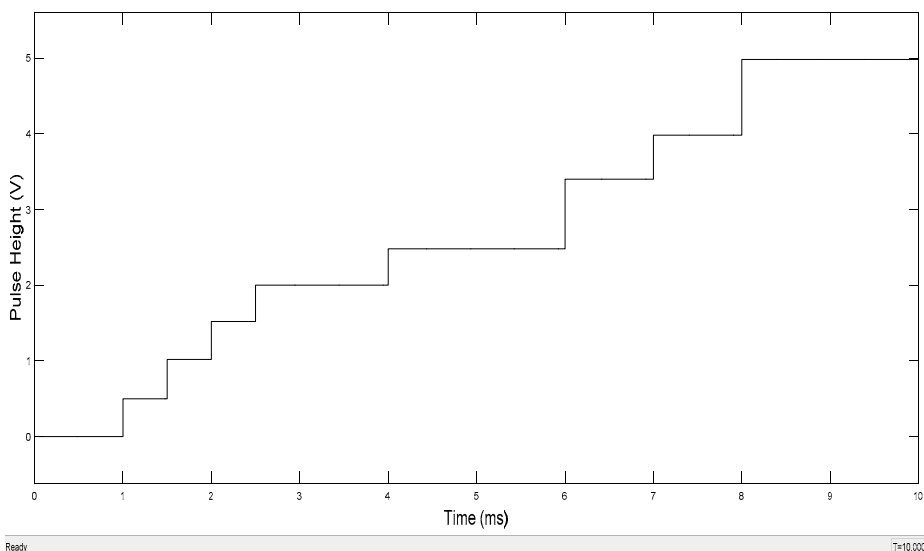

**Figure 2.** Typical staircase sequence signal $x(t)$.

### 2.2. Amplifier Output Representation

For the signal represented in Equation (1) through the CSA, the output signal of CSA can be written as:

$$x_{amp} = \frac{Q}{C_f} e^{-\frac{t}{\tau}} u(t) \tag{3}$$

which is a negative exponential signal, and the Laplace transform of Equation (3) is:

$$X_{amp}(s) = \frac{Q}{C_f} \frac{1}{s + 1/\tau} \tag{4}$$

where:

$Q$ represents the detector output charges;

$C_f$ is the CSA feedback capacitance;

$\tau$ is the CSA time constant. This is equal to $C_f R_f$ and, if it is too small, the output of the CSA will decay rapidly. Similarly, if it is too large, the output of the CSA will decay very slowly. In order to reduce the negative exponential signal pile-up, in theory, the time constant of the CSA should be minimal. In this case, the possibility of pile-up is small. However, this is difficult to realize in practical circuits if the time constant is too small.

In order to facilitate the simulation discussion, we assume that $Q = C_f = 1$; then, the formula can be expressed as:

$$X_{amp}(s) = \frac{1}{s + 1/\tau} \tag{5}$$

The corresponding Equation (5) of the z-transform form is:

$$X_{amp}(z) = \frac{1}{1 - dz^{-1}} \tag{6}$$

where $d = e^{-T_s/\tau}$. In the following discussion, we assume that $T_s = 0.01s$, $\tau = \frac{1}{5}s$, and $d = e^{-5T_s} = e^{-0.05} = 0.95$; then, Equation (6) can be expressed as:

$$X_{amp}(z) = \frac{1}{1 - 0.95z^{-1}} \tag{7}$$

Figure 3 represents the negative exponential output signal of the CSA when the input signal is the step signal.

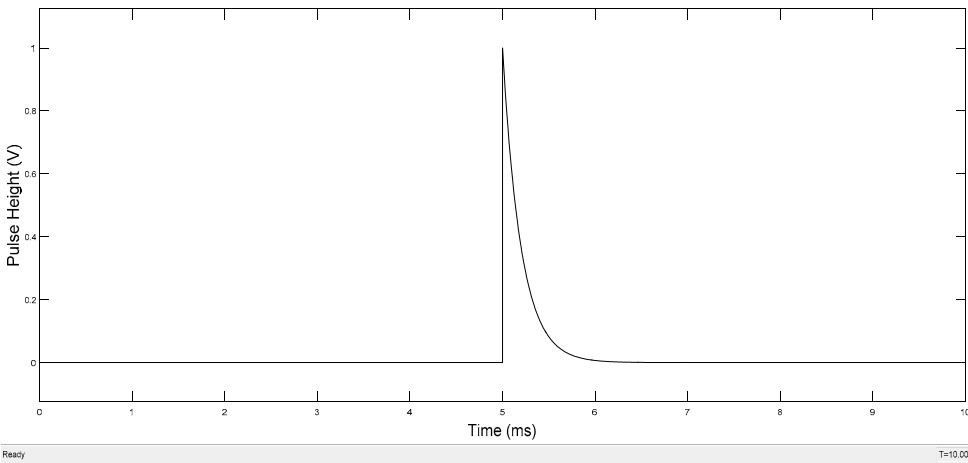

**Figure 3.** The typical CSA output.

### 2.3. Trapezoidal Pulse Shaping Algorithm

The trapezoidal pulse shaping (TPS) algorithm shows that when the input is a single discrete time negative exponential signal, using the TPS algorithm, the output becomes

the single discrete time trapezoidal pulse. Now, we discuss the issue in three steps. The first step is the trapezoidal pulse synthesis, as shown in Figure 4, which is conducted through the $x_1$ operation, including the time shift and reverse, and summation to obtain the trapezoidal shaping output $x_0$, as expressed as Equation (8):

$$x_o = \sum_{i=1}^{4} x_i \tag{8}$$

where $x_1 = (V_{\max}/t_a)tu(t)$, $x_2 = -x_1(t - t_a)u(t - t_a)$, $x_3 = -x_1(t - t_b)u(t - t_b)$, and $x_4 = -x_1(t - t_c)u(t - t_c)$, $t_a$ is the rising time of the trapezoidal pulse, $t_b - t_a$ is the duration of the flat top, $t_c$ is the total width of the trapezoidal pulse, and $V_{\max}$ is the height of the trapezoidal pulse. The second step involves using the z-transform, and Equation (8) can be expressed as:

$$X_o(z) = \frac{(1 - dz^{-1})(1 - z^{-n_a})(1 - z^{-n_b})z^{-1}}{n_a(1 - z^{-1})^2} \tag{9}$$

where $n_a = t_a/T_s$, $n_b = t_b/T_s$, and $n_c = t_c/T_s$, and $T_s$ is the sampling rate of the ADC. The third step is as follows—according to Equations (6) and (9), Equation (10) can be expressed as:

$$H(z) = \frac{X_o(z)}{X_i(z)} = z^{-1}\frac{(1 - az^{-1})}{n_a(1 - z^{-1})}\frac{1 - z^{-n_a} - z^{-n_b} + z^{-n_c}}{1 - z^{-1}} \tag{10}$$

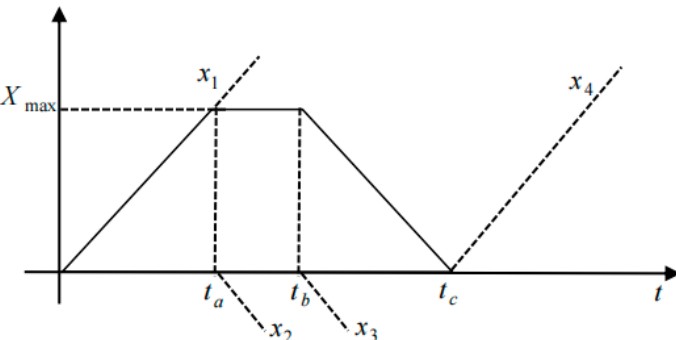

**Figure 4.** Trapezoidal pulse synthesis.

This is the known expression for the trapezoidal shaping algorithm. In the following discussions, we assume that:

$t_a = 0.05s$, $t_b = 0.15s$, $t_c = 0.2s$, $T_s = 0.01s$, $d = e^{-5T_s} = e^{-0.05} = 0.95$, $n_a = 0.05/0.01 = 5$, $n_b = 0.15/0.01 = 15$, $n_c = 0.2/0.01 = 20$.

$$H(z) = z^{-1}\frac{(1 - 0.95z^{-1})}{5(1 - z^{-1})}\frac{1 - z^{-5} - z^{-15} + z^{-20}}{1 - z^{-1}} \tag{11}$$

Assuming that the detector output is the single step signal, as shown in Figure 1, then the CSA output is the standard negative exponential signal, which can simulate the real nuclear signal, as shown in Figure 3. Using the TPS algorithm processing, the output of the trapezoidal signal is shown in Figure 5. The No1 pulse is the negative exponential signal and the No2 pulse is the trapezoidal shaping pulse. This has several advantages when compared with the traditional analog CR-RC4 shaping because it has better energy resolution. However, when the input has a staircase sequence signal, the output of the CSA shows the sequence of an added negative exponential signal; then, the trapezoidal output sequence will be preceded by Equation (10) and will result in greater complexity. Specifically, when there is a short time interval between two signals, the pile-up will happen almost immediately. This is discussed in further detail.

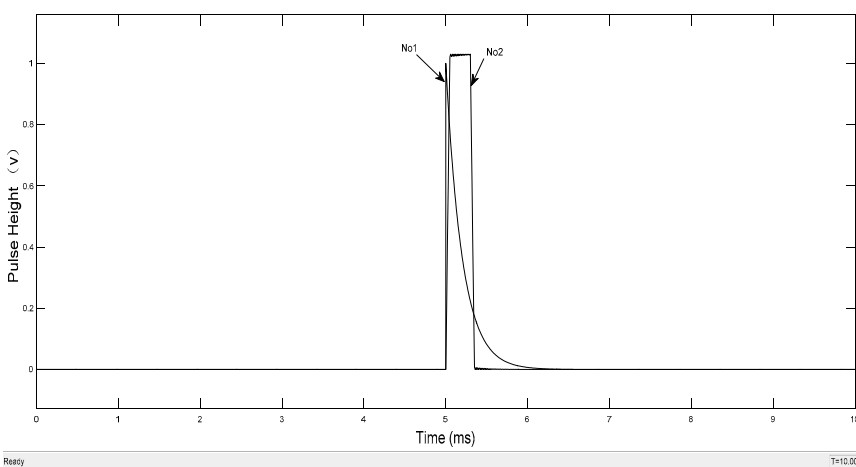

**Figure 5.** Trapezoidal pulse output.

## 3. Discussion of the Pile-Up Pulses

### 3.1. No Pile-Up Using TPS Algorithm

As shown in Figure 6, in this case, the time interval of the two adjacent arbitrary staircase input signals has enough width and, at the same time, the time constant of the CSA is shorter. The negative exponential output of the CSA does not exist in the pile-up, which is proceeded by Equation (7), and the output sequence of the trapezoidal shaping does not include pile-up, as shown in Figure 7. This is the ideal result. However, in practice, the time interval will change from us to ms and is accompanied by a difference in the ionizing position.

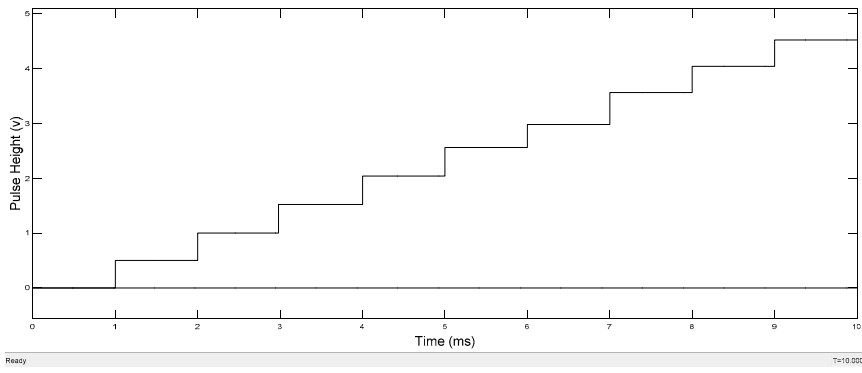

**Figure 6.** Detector equivalent outputs without pile-up.

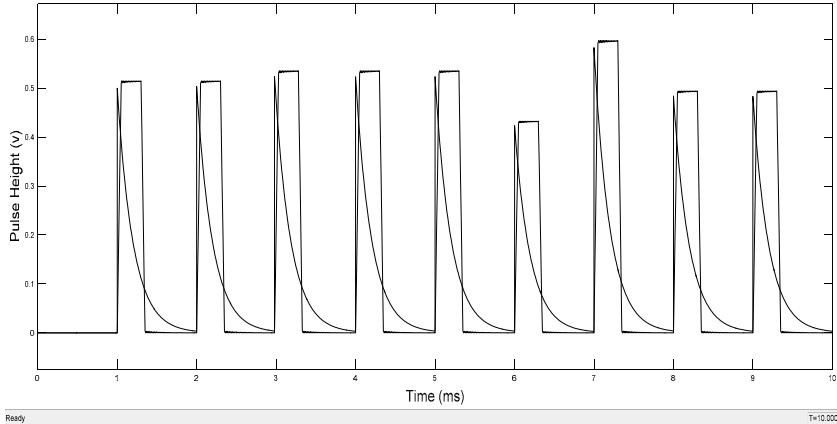

**Figure 7.** Non-pile-up using the TPS algorithm.

### 3.2. Pile-Up 1 Using the TPS Algorithm

As shown in Figure 8, the No1 pulse is closer to the No2 pulse. In this case, the pile-ups of the two negative exponential outputs for the CSA signals are proceeded by Equation (7), as shown in Figure 9. The No1 negative exponential pulse tails are softly added to the No 2 negative exponential signal leading edge, but this is not a serious issue. Hence, the sequence output of the trapezoidal shaping is preceded by Equation (11), as shown in Figure 9, which indicates the No3 and No4 pulses. This does not decrease the energy resolution; that is, it is the most effective pulse. However, if it is preceded by the traditional CR-RC4 shaping, the two negative pulses, i.e., the No1 and No2 pulses, are unavailable. This means that the energy resolution will worsen, and suggests that the trapezoidal shaping is better than the CR-RC4 shaping.

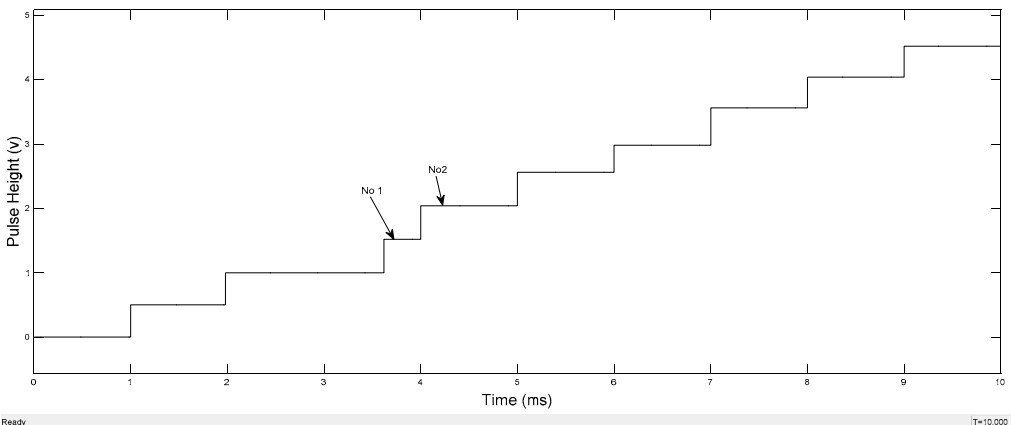

**Figure 8.** Detector equivalent outputs corresponding to pile-up 1.

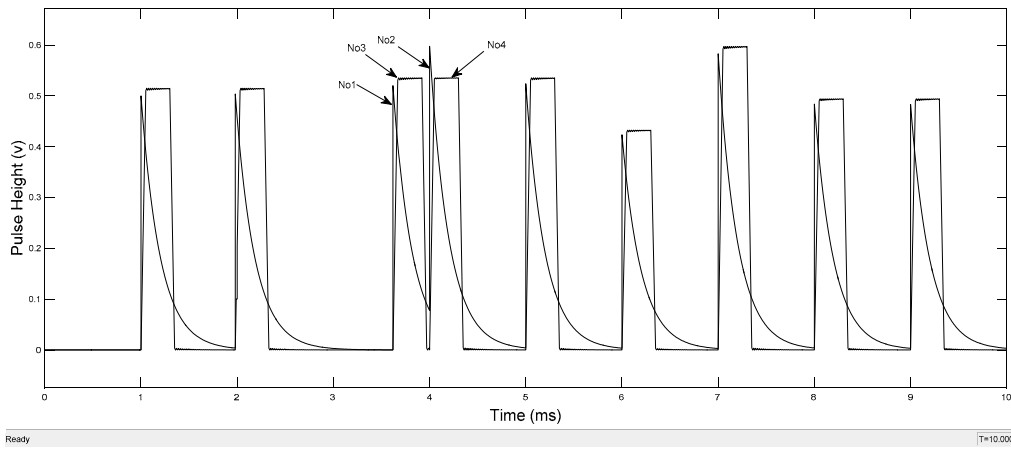

**Figure 9.** Pile-up 1 using the TPS algorithm.

### 3.3. Pile-Up 2 Using the TPS Algorithm

Another serious case is shown in Figure 10, where the No1 pulse is much closer to the No2 pulse. Hence, the two negative exponential output signals of the CAS become closer and the pile-up is preceded by Equation (7); at the same time, the pile-up of the output of the trapezoidal shaping is proceeded by Equation (11). Although the two trapezoidal pulses can be linked to become one convex profile (the No3 pulse in Figure 11), the pulse will be ineffective, and the adjacent pulses (the No1 and No2 pulses in Figure 11) cannot be separated completely. To account for the energy spectrum, we ignore the two pulses, resulting in worse resolution and inaccurate measurement. In the next section, the new method is used to address this problem.

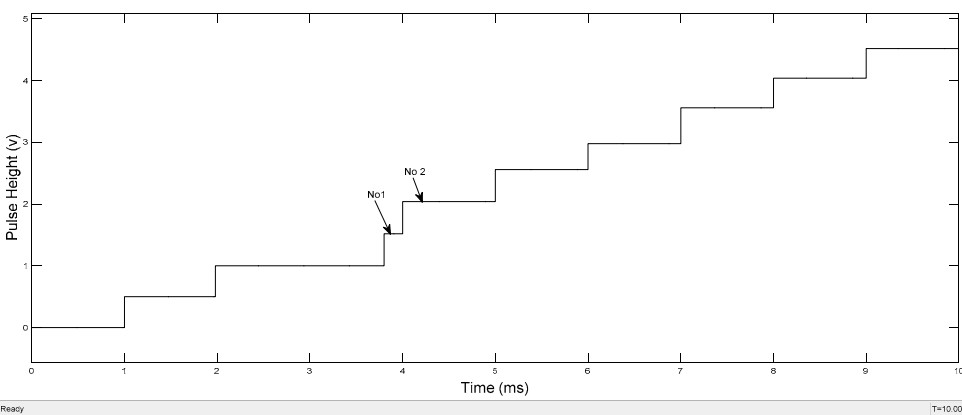

**Figure 10.** Detector equivalent outputs corresponding to pile-up 2.

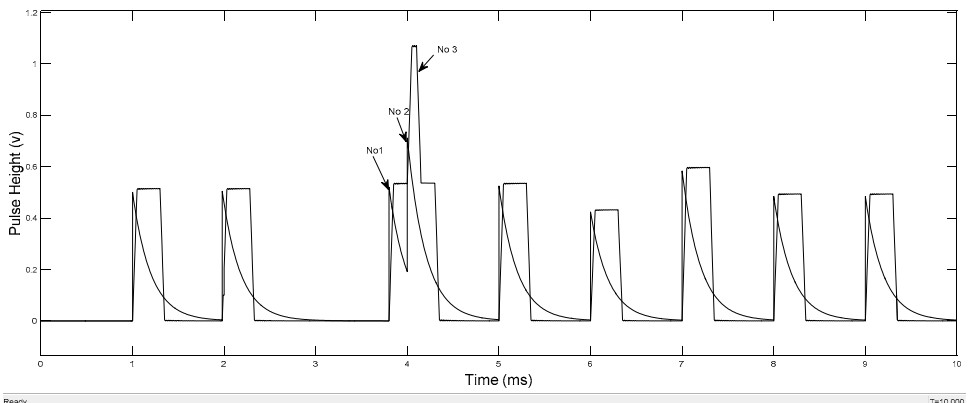

**Figure 11.** Pile-up 2 using the TPS algorithm.

## 4. Improved Trapezoidal Shaping Algorithm

### 4.1. Improved Trapezoidal Shaping Algorithm

From the above analysis, it is observable that pile-up 2 is a serious problem in energy spectrum accounting, and definitely affects the energy resolution and measurement. In order to attain the better energy resolution, a new function was built. This is now discussed in detail.

We rewrite the trapezoidal pulse shaping algorithm as:

$$H_1(z) = \frac{(1 - dz^{-1})(1 - z^{-n_a})(1 - z^{-n_b})z^{-1}}{n_a(1 - z^{-1})^2} \tag{12}$$

We also add another factor:

$$H_2(z) = \frac{z - 1}{z} \tag{13}$$

This is a differential factor, which differentiates the trapezoidal pulse, the rising edge, and falling edge of the trapezoidal pulse. These comprise the flat part of the trapezoidal pulse, which becomes zero, and the flat part after differentiation changes in positive proportion to the flat part of the original trapezoidal pulse.

Then:

$$H_{final} = H_1(z)H_2(z) = \frac{(1 - dz^{-1})(1 - z^{-n_a})(1 - z^{-n_b})z^{-1}}{n_a(1 - z^{-1})^2} \frac{z - 1}{z} \tag{14}$$

By substituting the parameters of Equation (11) into Equation (14), we obtain Equation (15), which is the new trapezoidal pulse shaping (NTPS) algorithm:

$$H_{final} = \frac{(1 - 0.95z^{-1})(1 - z^{-5})(1 - z^{-15})z^{-1}}{5(1 - z^{-1})^2} \frac{z - 1}{z} \tag{15}$$

This is discussed in detail.

Figure 12 is the Bode diagram of Equation (15) or the NTPS algorithm. It shows the special low pass filter with the low frequency domain and the sharp attenuation out of the cut-off frequency. Figure 13 is the map of the pole-zero. The zero and pole position can be changed to adjust for the transform function to achieve a better SNR and better shaping in practice.

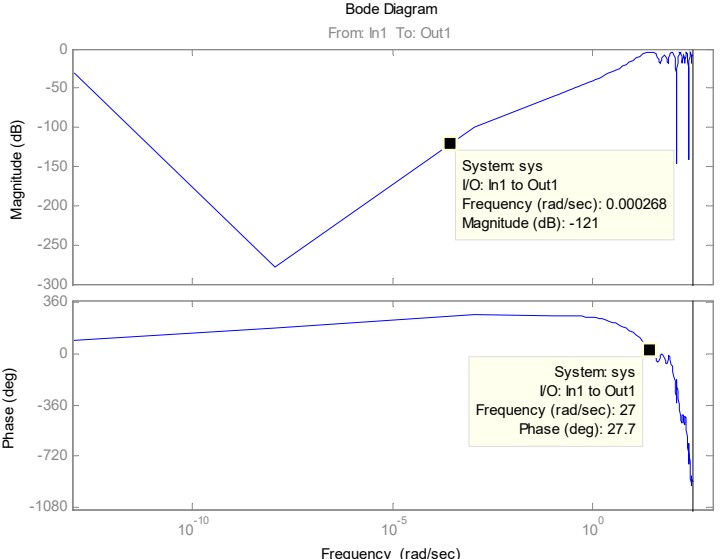

**Figure 12.** Bode diagram of Equation (15) or the NTPS algorithm.

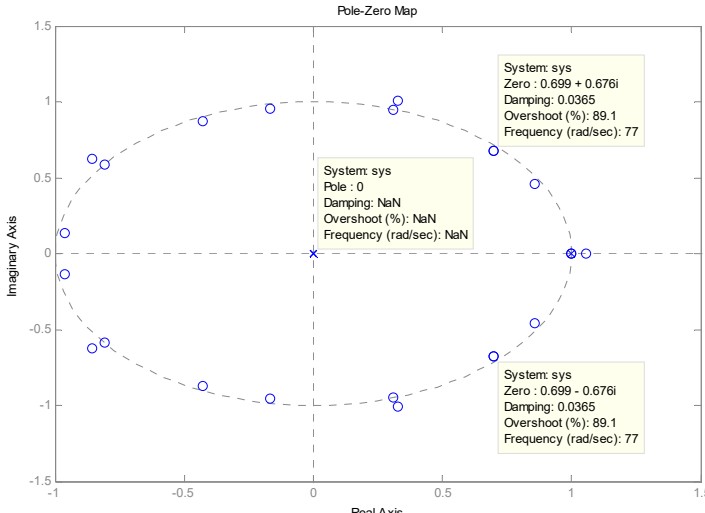

**Figure 13.** Pole-zero map of the Equation (15) or the NTPS algorithm.

Figure 14 shows the Simulink diagram of the NTPS algorithm. The simulation is easily implemented by MATLAB using the NTPS algorithm to process the negative exponential signal. The parameters are the same as those in Equations (11) and (15).

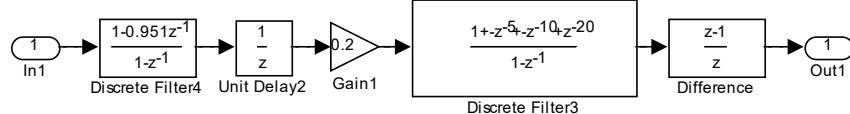

**Figure 14.** The NTPS algorithm Simulink diagram.

We discovered that the use of the NTPS algorithm can be used to influence the input negative exponential signal sequence of the CSA; regardless of whether the pile-up happened or not, proceed using the TPS algorithm; if pile-up happened, proceed using the NPTS algorithm. Next, we discuss two pile-up cases.

*4.2. Pile-Up 1 Processing Using the NTPS Algorithm*

As shown in Figure 15, there is pile-up between the No1 and No2 negative exponential pulses; using the TPS algorithm represented in Equation (10), the pile-up still exists as shown in Figure 8. The No3 and No4 trapezoidal pulses, however, still experience pile-up when using the NTPS algorithm, which is represented in Equation (14). By comparing the two results, the two pulses are separated successfully by the NTPS algorithm.

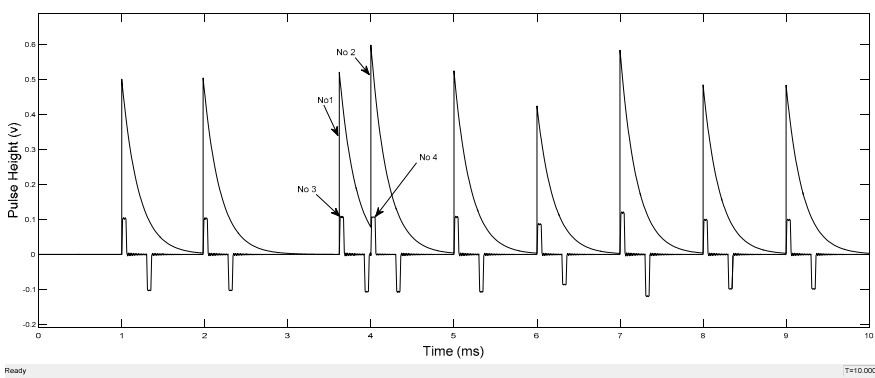

**Figure 15.** Pile-up 1 using the NTPS algorithm.

*4.3. Pile-Up 2 Processing Using the NTPS Algorithm*

As shown in Figure 16, the No1 and No2 pulses represent the negative exponential output sequence of the CSA. For the No1 and No2 pulses, when the pile-up negative exponential is preceded by the TPS algorithm represented in Equation (10), the output trapezoidal pulses will still pile-up because of the small time interval between the No1 and No2 pulses, as shown in Figure 11. This implies that the TPS algorithm is not effective. However, if the negative exponential output sequence of the CSA is preceded by the NTPS algorithm, which is denoted in Equation (14), the pile-up pulses will be successfully separated, as shown in the No3 and No4 pulses in Figure 16. This suggests that the energy will be greatly improved.

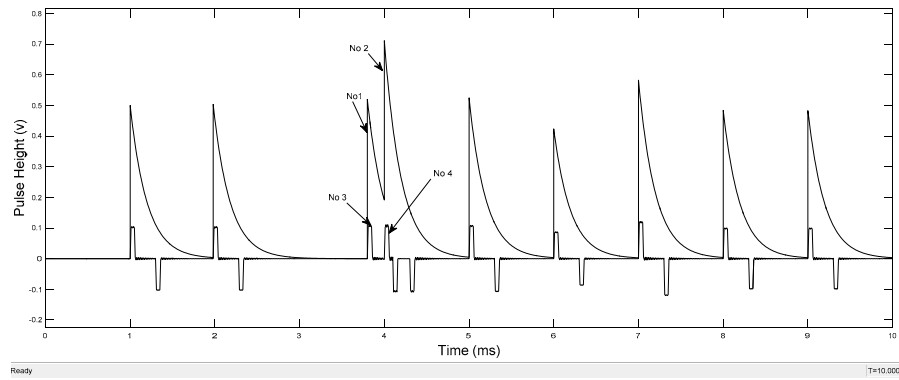

**Figure 16.** Pile-up 2 using the NTPS algorithm.

## 5. Comparison of Two Algorithms on Energy Resolution

As shown in Figure 17, 5.3% of the energy resolution of the TPS algorithm is represented by the blue mark "o", and 4.2% of the energy resolution of the NTPS algorithm is represented by the blue mark "*". The simulated photoelectric peak appears in the 47th channel. Obviously, the energy resolution of the NTPS algorithm is better than that of the TPS algorithm. In the simulation, we used the Signal Builder module in Simulink to establish the input signals. After TPS and NTPS algorithm processing, the data were saved in Excel. MATLAB was used to write a multi-channel amplitude analyzer program to process these data.

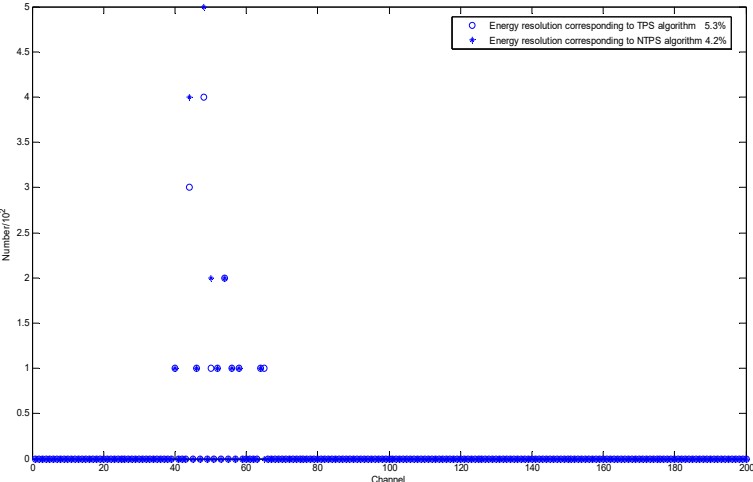

**Figure 17.** Comparison of the two algorithms in terms of energy resolution.

## 6. Conclusions

In this paper, the recursive algorithm of the trapezoidal pulse shaping is derived from the z-transform method. Adding a sub-function, we invented a new formula to process the pile-up. In theory, the NTPS algorithm can process any TPS algorithm pile-up. This is an effective approach that determines the total number in the spectrum and is more easily implemented by the DSP processor. In addition, the energy resolution of the NTPS algorithm is better than that of the TPS algorithm.

**Author Contributions:** Conceptualization, J.W.; methodology, X.T. All authors have read and agreed to the published version of the manuscript.

**Funding:** This research was funded by Fundamental Science on Nuclear Waste and Environment Security Laboratory, Southwest University of Science and Technology, grant number Grant no.15YYHK17.

**Acknowledgments:** We would like to thank Quanwei Li, who carefully read the manuscript and gave valuable advice.

**Conflicts of Interest:** The authors declare no conflict of interest.

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
