# Peer review of "Improvement in Trapezoidal Pulse Shaping Pile-Up in Nuclear Signal Processing"

_electronics, doi:10.3390/electronics11111745_

Round 1
Reviewer 1 Report
The paper presents a discussion of the trapezoid pulse shaping technique, widely used in contemporary digital signal processing. It also proposes an improvement which should help reducing pile-up effects and enhance the energy resolution.
The approach to the issue is very well organised and logically consequential, starting from a discussion of known literature and eventually presenting (Chapt. 4) the improved algorithm with examples.
However, throughout the paper, detailed treatment and much less explained steps are alternated and somewhat unbalanced. A more uniform use of details should be adopted by the authors, especially when recalling the referenced articles.
The final part discussing the performance of the new algorithm should be more detailed, for example the role of the pole-zero in the tuning of the digital filter would deserve a deeper discussion. In addition, it would be of great help for the reader to see some exemplary energy spectra derived from this analysis. This would much better support the author's claim of an improved energy resolution and have a better awareness of the limits of the method.
As for other comments, I would advise the authors to better check the references, since in the text some of them are apparently mistaken. For instance, [1] is a work by Radeka, not Koeman, performed at Brookhaven National Laboratory (not Philips). At line 30, Hongxu et al. are referred to, but [4] is by Stein et al., as in the reference list.
Some improvement should also be considered for the style and English language, repetitions, typos (for example, use of 'the' instead of 'a' throughout the text, 'counting rates' instead of 'counter rates' in the abstract, 'Different' instead of 'Many' or 'Various', etc.)
The formulae at line 74 should be revised ( tau is unitless?; d=exp(-T/5)).
At line 83, the x1(t) is cited, which is defined later at line 86. The sequence should be inverted.
Figures' captions should be better expanded to cast more information to the reader. In addition, many figures need a revision of font size in order to be clearly displayed.
In summary, I believe the work presented is interesting and deserving a publication, but in my opinion it needs a revision in order to improve the overall style and the discussion of Chapt. 4 with the relevant conclusions on energy resolution.
Author Response
Dear reviewer,
I'm glad to receive your professional comments. According to your comments, I have revised some contents that I think need to be discussed. Thank you very much!
kind regards,
wujunlong

Reviewer 2 Report
This work attempts to find a method to separate pile-up issues when using the trapezoidal filter. In so doing, they create a function to adjust the filter and are able to identify pile-up situations more easily.
This is a good idea and can have good results but this is not displayed in the paper.
Issues to be addressed:
1) the H2(z) function in equation 13 should be explained more, where does it comes from and why was it chosen to be the function to implement. In other words, the reasoning for this function should be explained and compared to other functions that have been implemented since the last used reference of 2003 (reference 5).
2) It should be shown clearly the effect of the additional function (equation 15) as a function of the time difference between the pileup signals to see where this function breaks down.
3) There is also no study on the applicability of this function for different waveform decay times or how the resolution changes for the trapezoidal filter output with equation 15 vs equation 11. Without such a study, the advantage of using a trapezoidal (of better energy resolution) has not been verified. It could be that the NTPS can pick out the pile-up but if the trapezoidal filter does not do the job of energy resolution in the first place, then the purpose is defeated.
These are a list of three essential parts to make this a complete study. There may be more but as it is this paper is don't scientifically sound in its conclusion.
Besides this, there are several grammatical errors, formatting issues, and use of the wrong structure in paragraphs. Two pages (page 5 and page 7) of the paper are written with text in between figures and it is not clear if the text is a caption for the figures or part of the text of the paper. The paper needs a complete overhaul.

Author Response

(The authors gave the same response as above.)

Reviewer 3 Report
In the presented paper, Authors propose a new and easy to implement recursive algorithm to improve the anti-pile-up ability.
The manuscript is well written and prepared. Easy to read and follow. I recommend accepting the manuscript for publication after major revision.
Introduction need to be re-written, the flow of introduction is poor. Would you explicitly specify the novelty of your work? What progress against the most recent state-of-the-art similar studies was made? Please carry out a more detailed literature analysis.
More literature references are needed. Need more latest references.
Author Response
Dear reviewer,
I'm glad to receive your professional comments. According to your comments, I have revised some contents. Thank you very much!
kind regards,
wujunlong

Round 2
Reviewer 1 Report
Let me first acknowledge that the Authors have apparently done some effort to improve the clarity of the paper. The Authors have better outlined some details of the method and the results of their research. This is very appreciable.
However, I am still doubtful about some aspects of the discussion.
As selected examples:
the recurrent use of the undefined term CAS together with the term CSA, which is instead clearly defined, should deserve some order;
the terms 'spectrometer', 'ionization chamber', never defined or mentioned as a reference set-up the reader should be aware of;
several repetitions are still found in the text;
some claims (energy resolution improvement) not sufficiently supported.
The overall feeling for the reader is he must make an unnecessary effort to put things together and get a clue about it.
This feeling does not reflect the value of the research, which instead would deserve a clearer and polished presentation.
The concept of "energy resolution" is introduced in the abstract and recalled in some crucial parts of the work, but I think that an exemplary energy spectrum would make the Author's claimed improvements straightforward for the reader. I believe that pulse-height versus time of many figures is very useful to illustrate the mechanism of the algorithm, however does not say much about the energy resolution.
The latter could easily be demonstrated by transforming the time dependent data into time-integrated pulse-height spectra, showing the effect of the improved pile-up rejection, e.g. simulating the photo-peak of some gamma emitter, or the energy peak of some X-ray calibration source, like the commonly used Fe-55, in a detector, and quoting deltaE/E.
This makes me more confused when I consider that the Authors' newly proposed algorithm, based on a sort of trapezoid differentiation, through the introduction of the factor H2(z)=(z-1)/z, appears as an improvement of time resolution, rather than of energy resolution.
I must admit that, despite the appreciable effort done to address previous comments in Report 1 and improve the presentation, my personal advise is to consider the work not sufficiently ready for publication yet.
In other terms, the content is valuable, and I am happy that the NTPS method deserved a patent, as reported by the Authors in the ancillary materials, but the paper should need some extra polishing to target the general reader.
Besides that, I must also notice that some of my previous comments from Report 1 have been overreacted. In particular the full substitution of the article 'a' with 'the'.
I might have been little clear myself, but I did not expect a replacement everywhere. I just advised/asked to check thoroughly for such English language issues. In this respect, I would encourage the Authors to seek for a native English speaker, or a Colleague very experienced in English language, in order to get a language and style review before any further submission.
I am still convinced that this research deserves a publication, but
my final recommendation is that the paper still needs improvements to get published.
Author Response
Dear reviewer:
According to your suggestion, the article has been revised.
Kind regards.
wujunlong.

Reviewer 2 Report
From the corrections/edits that have been turned in by the authors, I see several errors and ommissions. My largest concern is that this work is not based on a complete literature search. Currently, several major laboratory facilities that use digital electronics as well as the large manufacturers of digital electronics include pileup rejection in their pre-PHA algorithm. There is little or no credit or comparison given to all the work that exists on pile-up rejection. For an example, one should look at the XIA manuals for PIXIE and well as the CAEN digitizers, amond others.
After the last review, there was a current reference added to the reference list but it appears the author has not looked at the content or the references therein.
The authors, I am sure understand that the exponential decay pulses that they use as the signals in this work are going to be different for different detectors and vary with different energies. The Tau will be different in each case. The pileup rejection or the pile-up tag can then be used to either reject the two pulses or determine the energy of the pulses that were piled up. The difference between the two comes from the fact that the tau and risetime of the pulses are different and one can then make a comparison of where the breakdown between tau and rise time happen to make a criteria for pulse rejection or PHA using the trapizoidal filter on the piled up pulses. There is no work to indicate this or discuss the broad applicapility of this.
Lastly, there are several gramatical errors and wrong use of "the" article where "a" should be used. There is not enough care taken in writing this article in the english language.
In summary, for work that is based on simulated signals, there is very little that has been done to analyse the effect of the pile-up determination, to determine the extent of its use and where it breaks down. This is not a new idea, there are publised articles in nature and PRL thak have used pileup to seperate out alpha energies as well as Gamma energies (two very different time-constant signals). There has been little to know literature search to see what prior work has been done in regard to pileup rejection in conjunction with the trapeziodal filter. Gramatical errors are still rampant in this work.
Author Response

(The authors gave the same response as above.)

Reviewer 3 Report
After correction I recommend the manuscript for publication.
Author Response
Dear reviewer:
According to your suggestion, the article has been revised. Thank you.
Kind regards.
wujunlong.
Round 3
Reviewer 1 Report
The relevant doubts raised in my previous report appear to be answered. I can not still understand why the Authors use CSA and CAS. The latter is not defined, unless it is only a typo. The Editors should address it, together with some very light English polishing.

Author Response
Dear reviewer:
According to your suggestion , the article has been revised.
Kind regards.
wujunlong.

Reviewer 2 Report
As mentioned earlier, this work is not unique. Using the differential of the input waveform to determine the pileup in digital electronics is used widely. This work continues to assume their work is unique and has not done sufficient background literature search to learn about what has already been done. For example, after doing a simple search, I have found a peer review journal article that uses the differential method described in this article for pile up identification and it appears they have experimental data from their algorithm that does a very good job with the pileup identification and the extraction of energy information from the associated trapezoidal filter. See:
1)https://www.sciencedirect.com/science/article/abs/pii/S0969804300002475?via%3Dihub
2) https://iopscience.iop.org/article/10.1088/1674-1137/39/6/068201/meta
3)https://www.sciencedirect.com/science/article/abs/pii/S0168900217302243
The third article uses precisely the same method described in this work.
I reiterate my previous concerns about the authors needing to be more diligent on what work has been done prior to their wanting to publish this work as new. A thorough literature search needs to be done to confirm their work is unique. Not only this but the authors need to show how their work is unique and robust compared to other work that has used the waveform differential to identify or reject pileup and have tested this by implementing their algorithm within their FPGA and shown the pileup analysis in real data (not just simulated). If there was no other work or implementation of pileup analysis in digital electronics then this would be a good article to show a new method. However, this is not the case today. Because other work does exist what the authors show in their simulation appears to be not unique and needs further justification to show that it is not a repeat find and that there is more practical value to their work compared to others.
Author Response

(The authors gave the same response as above.)
